# Bipolar Membrane Electrodialysis for Sulfate Recycling in the Metallurgical Industries

**DOI:** 10.3390/membranes11090718

**Published:** 2021-09-18

**Authors:** Wouter Dirk Badenhorst, Pertti Kauranen, Heikki Pajari, Ronja Ruismäki, Petri Mannela, Lasse Murtomäki

**Affiliations:** 1Department of Chemistry and Materials Science, School of Chemical Engineering, Aalto University, P.O. Box 1600, 02150 Espoo, Finland; kuldeep.kuldeep@aalto.fi (K.); wouter.badenhorst@aalto.fi (W.D.B.); 2School of Energy Systems, LUT University, P.O. Box 20, 53851 Lappeenranta, Finland; pertti.kauranen@lut.fi; 3VTT Technical Research Center of Finland, Tietotie 4E, 02150 Espoo, Finland; heikki.pajari@vtt.fi (H.P.); Petri.Mannela@vtt.fi (P.M.); 4Finnish Minerals Group, Keskuskatu 5B, P.O. Box 800, 00101 Helsinki, Finland; ronja.ruismaki@mineralsgroup.fi; 5Department of Chemical and Metallurgical Engineering, School of Chemical Engineering, Aalto University, 02150 Espoo, Finland

**Keywords:** bipolar membranes electrodialysis, ion exchange membrane, sodium sulfate, sodium hydroxide, sulfuric acid, electro-transport

## Abstract

Demand for nickel and cobalt sulfate is expected to increase due to the rapidly growing Li-battery industry needed for the electrification of automobiles. This has led to an increase in the production of sodium sulfate as a waste effluent that needs to be processed to meet discharge guidelines. Using bipolar membrane electrodialysis (BPED), acids and bases can be effectively produced from corresponding salts found in these waste effluents. However, the efficiency and environmental sustainability of the overall BPED process depends upon several factors, including the properties of the ion exchange membranes employed, effluent type, and temperature which affects the viscosity and conductivity of feed effluent, and the overpotentials. This work focuses on the recycling of Na_2_SO_4_ rich waste effluent, through a feed and bleed BPED process. A high ion-exchange capacity and ionic conductivity with excellent stability up to 41 °C is observed during the proposed BPED process, with this temperature increase also leading to improved current efficiency. Five and ten repeating units were tested to determine the effect on BPED stack performance, as well as the effect of temperature and current density on the stack voltage and current efficiency. Furthermore, the concentration and maximum purity (>96.5%) of the products were determined. Using the experimental data, both the capital expense (CAPEX) and operating expense (OPEX) for a theoretical plant capacity of 100 m^3^ h^−1^ of Na_2_SO_4_ at 110 g L^−1^ was calculated, yielding CAPEX values of 20 M EUR, and OPEX at 14.2 M EUR/year with a payback time of 11 years, however, the payback time is sensitive to chemical and electricity prices.

## 1. Introduction

The mining and metallurgical industries produce large amounts of waste during the processing and neutralization of acidic streams [1]. Sulfuric acid is commonly used for leaching metals and their oxides [2,3] and sodium hydroxide is subsequently used for their precipitation and acid neutralization, as well as for pH control of other processes [4]. As a result, undesirable sodium sulfate (Na_2_SO_4_) solution is produced [5]. The rapidly growing battery industry is one of the sectors in which such solutions are produced. For example, typical production processes for nickel, manganese and cobalt (NMC) battery cathode materials comprising of leaching and co-precipitation stages produce liquid streams rich in Na_2_SO_4_ [6,7,8]_._ The stream can be considered as a potential source for by-product(s) if revenue can be generated from it [9], or a waste stream if discharged or disposed, with approximately half of the world production of commercially traded Na_2_SO_4_ generated as a by-product [10].

Na_2_SO_4_ is not a direct threat to the environment, as sulfate is a chemically inert, non-volatile and nontoxic compound [11]. On the other hand, despite being one of the key nutrients [12], sulfate increases the salinity level of surface water [13], affecting the cycling of other nutrients, binding of metals, and formation of toxic substances in aquatic systems [12]. In high levels, it is harmful to freshwater biota. However, the contribution of sulfate-rich emissions from anthropogenic activities is poorly known [12]. In European water legislation, sulfate is not listed as a polluting substance to be taken into account for setting emission limit values [12,14]. Na_2_SO_4_ can be crystallized and, to a certain extent, sold or diluted and discharged into waterways, provided that it complies with environmental legislation. However, discharging of sulfate waste streams should preferably be controlled and monitored systematically [12] or avoided when possible [15], although not strictly regulated in most countries. Due to increasing pressure related to water sustainability issues [16], it is expected that environmental regulations for wastewater treatment could become stricter. Should regulations become stricter, the processing of Na_2_SO_4_ effluents could become a requirement despite not having direct financial incentives.

Modifications in processes may eliminate the formation of Na_2_SO_4_ emissions [7]. If this is not an option, a variety of options for the valorization of Na_2_SO_4_ exist. Van der Bruggen et al. [17] explored whether thermal processes, such as evaporation and distillation, can be suitable for the reduction in the waste fraction of brine. Atia et al. recovered Na_2_SO_4_ with a purity of 96% by evaporation–crystallization. Sodium sulfate meeting commercial specifications could be used as a raw material in, for example, the detergent or pulp and paper industry [10]. However, these methods are not as applicable if the salinity of the effluent is below 5000 ppm [18]. In cases where the salinity is lower than that, more commonly membrane separation processes are being employed, such as electrodialysis (ED), bipolar membrane electrodialysis (BPED) and reverse osmosis (RO). [18] These techniques have long been utilized for desalination, softening, and contaminant removal from waste effluents. Except for RO which relies on porous membranes, both ED and BPED processes rely on cation and anion exchange membranes to selectively transport ions [11,19,20,21] upon application of an electric voltage over an ED or BPED stack, of which multiple configurations are shown in Figure 1. Thus, a combination of different configurations of ion exchange membranes allows either dilution of waste salt streams or recovery into corresponding acid and base. The recovery of acid and base from its corresponding salt solution is achieved through bipolar membranes splitting water into protons and hydroxides [15]. Therefore, in the BPED of Na_2_SO_4_, recombination of protons with sulfate ions, and hydroxide ions with sodium ions leads to the formation of H_2_SO_4_ and NaOH solutions [5]. Recovered acid and base solutions can be reused for leaching and precipitation of metals or in other process streams.

According to Bazinet and Geoffrey, the worldwide market for ED equipment reached USD 318 million in 2019 and is expected to grow to market values of USD 458 million with annual growth between 5.5–5.8% up to 2025 [22]. In addition to this, the number of companies providing ED equipment to market, totals 45 in 2020, from a mere 6 companies in 1980 [22]. The ED-based techniques are receiving increased attention for the treatment of waste water, as they offer promising prospects for the recovery of selected ions in the form of concentrated streams [23] or reuse valuable compounds from saline streams by using bipolar membranes [24]. Water dissociates into protons and hydroxyl ions at 0.83 V across the BPM, and combining them with salt constituents produces acid and base in the two product steams. Kroupa et al. (2014) have treated a Na_2_SO_4_ effluent coming from uranium processing through electrodialysis with bipolar membranes [25]. The aim of their lab-scale experiment was to study the effect of various process parameters on energy consumption, and to achieve the target concentration of 5.5 wt% (0.56 M) for H_2_SO_4_ and 6.5 wt% (1.63 M) for NaOH. The purity of the products was in the range of 71.9–83.3% for H_2_SO_4_ and 97.0–98.3% for NaOH. Kroupa et al. (2016) additionally investigated the BPED of sodium sulfate in four different stack configurations [26] and compared the results with the standard ED process. After running BPED (BCA configuration Figure 1a) of 35 g/L Na_2_SO_4_ (0.25 M) solution for 8 h, acid concentration of 60 g/L (0.6122 M) and base concentration of 46 g/L (1.15 M) were reached. From ED of either 25 g/L H_2_SO_4_ (0.255 M) or 25 g/L NaOH (0.625 M) solution in a three-compartment configuration (Figure 1b), acid concentration of 105 g/L (1.07 M) and base concentration of 98 g/L (2.45 M) were reached in 16 h. The same feed solution was used also in a four-compartment configuration (BCAA and BCCA configuration Figure 1c,d), but no significant effect was noticed on the acid concentration. However, the NaOH concentration was raised to 58 g/L (1.45 M) in the four-compartment configuration (BCCA configuration, Figure 1d). Kinčl et al. (2017) carried out BPED of pure Na_2_SO_4_ solution at pilot scale using heterogeneous membranes [27]. Their aim was to scale up their pilot trials results to an industrial scale. They achieved maximum product concentrations were 1.5% (0.16 M) for H_2_SO_4_ and 4% (1.0 M) for NaOH at a current density of 350 A m^−2^. The purity of their products was ca. 85% with the current efficiency of 62%. A recent experimental study was carried out by Bruinsma et al. (2021) [5] concerning BPED of (0.7 M) Na_2_SO_4_ in combination with reverse osmosis (RO). They operated BPED in BCA configuration (Figure 1a) with 10 repeat units (RUs) in a batch mode at a constant voltage of 30 V. Both the current density and salt splitting were observed 25% higher on increasing the temperature of the BPED from 25 to 35 °C. In contrast, cation exchange membrane (CEM) current efficiency is drastically dropped from 100% to 83% due to an increase in proton leakage. Furthermore, they achieved purity values of 99% for NaOH and 90% for H_2_SO_4_.

The overall BPED efficiency strongly depends on the temperature which affects the ionic conductivity of ion-exchange membranes (IEM). The temperature has a strong effect on diffusion coefficients, as defined by the Stokes–Einstein equation [28,29], and via the viscosity, the fluid velocity. Furthermore, the selectivity and permeability of ions through IEMs can also be affected as the hydration of ions decreases with an increase in temperature. In general, however, increased ionic fluxes [30] as well as lower membrane resistances [31] are expected at higher temperatures, which improves the energy efficiency of BPED. Thus, the temperature study is crucial in defining the optimal performance of the BPED process, which is needed in the design of industrial plants, considering capital costs and energy consumption. Degradation of membranes in terms of selectivity of ions [32], capacity loss, and lifetime depletion of the ion exchange resin are the major drawbacks of higher temperature operations [33].

In the last few years, the performance, stability and operating window of AEMs, CEMs and BPMs have improved significantly. Older generations of membranes typically showed low stability, mainly in caustic and strong acidic environments [21], and produced low purity products due to low selectivity and relatively low current efficiencies. With newer homogeneous membranes such as those offered by SUEZ Water Technologies and Solutions [34], higher purity products can be produced at elevated temperatures and at higher current density. Hence, we tested these membranes in order to determine the commercial viability of the process.

In this present paper, BPED of an industrial effluent is carried out in a feed and bleed mode which can be utilized in scaling up to an industrial-sized system. The aim of the study is to check the performance of the SUEZ membranes in a BPED stack with 5 and 10 repeating units of a BCA (BPM-CEM-AEM) configuration at different temperatures and current densities. Furthermore, the current efficiency was evaluated as a function of temperature, current density, and product concentrations. The capital and operational expenditures of an industrial-sized system are also discussed briefly.

## 2. Materials and Methods

All experiments were conducted with the Suez laboratory/bench scale MkI ED stack with an active membrane area of 280 cm^2^. Different sets of experiments were carried out with 5 and 10 repeating units (Figure 2), and conductivities, inlet stack pressures and flow rates were monitored for all flows by Indumax CLS50D, Cerabar M PMC51, and Promag 53H (Endress Hauser, Germany). In addition, pH and temperature for the feed effluent were measured by Orbisint CPS11D Memosens (Endress Hauser, Germany). The anode consisted of platinum-coated titanium and the cathode was made of stainless steel. The properties of the SUEZ membranes used in the tests are listed in Table 1 and contain manufacturer specified data on the water uptake (WU) and ion exchange capacity (IEC) whereas the membrane fixed charge was calculated, Equation 1. The composition of the effluent is given inTable 2, and the conductivity of the effluent was determined to be 85 mS cm^−1^.
(1)$$c_{m}=\frac{C_{iex}\rho_{m}}{1+X}$$ where $c_{m}$ is the membrane fixed charge (M), $C_{iex}$ the ion exchange capacity (meq g^−1^), $\rho_{m}$ the density of a hydrated membrane and *X* its water content, whereas the BPM used was manufactured through lamination of CR61P and the treated AR103P membrane.

The experiments were conducted in a batch feed and bleed configuration, which is depicted in Figure 3. When the conductivity of either the acid or base reaches its target value, 20% of its volume is replaced with DI-distilled water. Similarly, 30% of the feed compartment is replaced with fresh feed solution after reaching the target conductivity. These steps were repeated continuously, with the final product concentrations being determined through titration and the purity of the products were assessed using ICP-OES (SFS-EN ISO 11885:2009).

## 3. Result and Discussion

In this section, BPED results are presented and discussed for three temperatures (24 °C, 34 °C, and 41 °C). Generally, membrane stability tends to decrease above these temperatures, and for that reason, it was decided to keep 41 °C as the maximum temperature. At these temperatures three current densities were tested (300, 400 and 500 A m^−2^), which were well below the limiting current density, with the current efficiency being determined for each counterion.

### 3.1. Stack Voltage

Figure 4a,b show the stack voltage variation with current density and temperature. Raising the current density raises the cell voltage because its major part comes from ohmic losses in the membranes and the solutions. It also accelerates mass transfer at the solution-membrane interface and electro-osmosis in the membrane and raises temperature via Joule heat. Stack voltage is found to increase linearly with the current density at a fixed temperature (Figure 4a), whereas it decreases on increasing temperature at a fixed current density (Figure 4b) due to the higher conductivity of the solutions and lower membrane resistance [35] Energy efficiency should be improved by operating at 41 °C, which can be maintained with heat exchangers, additionally retaining heat produced by the operation of the BPED stack. At 41 °C and 500 A m^−2^ the voltage per stack is 2.6 V, whereas Kinčl et al. [27] operated at 3 V per stack for optimal performance using heterogeneous membranes at 350 A m^−2^.

### 3.2. Current Efficiency

In this part, the effect of current density and temperature on current efficiency is discussed. Current efficiencies of acid and base production were calculated from Equations (2) and (3) given below:(2)$$\frac{\eta_{\mathrm{acid}}^{\mathrm{tot}}}{100\%}=\frac{r_{H^{+}}^{A}}{N_{\mathrm{RU}}\left(I/F\right)}=\frac{V_{r}^{A}c_{H^{+}}^{A}}{N_{\mathrm{RU}}\left(I/F\right)}$$
(3)$$\frac{\eta_{\mathrm{base}}^{\mathrm{tot}}}{100\%}=\frac{r_{{\mathrm{OH}}^{-}}^{B}}{N_{\mathrm{RU}}\left(I/F\right)}=\frac{V_{r}^{B}c_{{\mathrm{OH}}^{-}}^{B}}{N_{\mathrm{RU}}\left(I/F\right)}$$
where $r_{H^{+}}^{A}$ (mol s^−1^) and $r_{{\mathrm{OH}}^{-}}^{B}$ (mol s^−1^) are the accumulation rates of protons in acid and hydroxide ions in base, $N_{\mathrm{RU}}$ number of repeating units, I electric current (A), F Faraday constant (As mol^−1^), $V_{r}^{A}$ (L s^−1^) and $V_{r}^{B}$ (L s^−1^) volumetric flow rates of acid and base products, and $c_{H^{+}}^{A}$ proton (M) and $c_{{\mathrm{OH}}^{-}}^{B}$ hydroxide ion (M) concentrations in acid and base. The current efficiency of the acid and base production was studied at varying temperatures with a fixed current density of 300 A m^−2^; the results are shown in Figure 5. A noticeably higher current efficiency is found at 41 °C compared with 23 °C for both acid and base production. It is not clear why this should happen, as the process is run under constant current conditions, although increased conductivities [35,36] and water splitting rate [36] have an effect on the stack voltage.

The impact of current density on the current efficiency of both acid and base production was also determined (see Figure 6a,b). It appears that the current efficiency is not greatly affected by an increase in current density, which implies that the process is not under mass transport control. Mass transfer limitation can reduce the current efficiency through the onset of unwanted side-reactions.

Current efficiency was also investigated as a function of BPED products concentration (Figure 7). At low product concentrations current efficiencies above 85% are observed, and on increasing product concentrations efficiencies drop to 55% for H_2_SO_4_ and 70% for NaOH, indicating higher leakage of co-ions. The lower H_2_SO_4_ efficiency compared with NaOH indicates that the proton leakage through the AEM is significantly higher than the hydroxyl leakage through the CEM when the concentrations of protons and hydroxide ions are equal in the acid and base, respectively. Furthermore, the same conditions of products were conducted with pure Na_2_SO_4_ (0.5 M) solution as feed solution. Surprisingly, there were no significant differences noticed, although the concentration of Na_2_SO_4_ in waste effluent was 110 g L^−1^ (0.775 M), Table 2. The minor variations in current efficiency can possibly be attributed to the effluent impurities either slightly fouling the membrane and or other side reactions occurring. In addition, these current efficiency results are comparable with our single membranes simulations and experimental studies [37].

The previous experiments were conducted with 10 repeating units. A single test was made with a stack containing five repeating units only, to study the effect of the number of repeating units on current efficiency (Figure 8a,b). Theoretically, current leaking through the liquid manifolds could increase as the number of RUs increases. The current efficiency of base and acid is evaluated as a function of current density in both cases. The addition of five repeating units does not greatly affect the current efficiency. Hence, it appears that with the present cell design and implementation (5 vs. 10 RU) there is no significant current leakage. However, the current leakage effects may become more significant at a higher number of RU.

Purities of the products are defined as
(4)$$\mathrm{Purity}\;\mathrm{of}\;\mathrm{acid}=\frac{m_{H_{2}{\mathrm{SO}}_{4}}}{m_{H_{2}{\mathrm{SO}}_{4}}+m_{{\mathrm{Na}}_{2}{\mathrm{SO}}_{4},A}}\times 100\%$$
(5)$$\mathrm{Purity}\;\mathrm{of}\;\mathrm{base}=\frac{m_{\mathrm{NaOH}}}{m_{\mathrm{NaOH}}+m_{{\mathrm{Na}}_{2}{\mathrm{SO}}_{4},B}}\times 100\%$$
where $m_{H_{2}{\mathrm{SO}}_{4}}$ is the mass concentration (g L^−1^) of sulfuric acid and $m_{{\mathrm{Na}}_{2}{\mathrm{SO}}_{4},A}$ concentration of sodium sulfate in the acid product; $m_{\mathrm{NaOH}}$ is the mass concentration of sodium hydroxide and $m_{{\mathrm{Na}}_{2}{\mathrm{SO}}_{4},B}$ the mass concentration of sodium sulfate in the base product.

In our experiments, both H_2_SO_4_ and NaOH are produced with a purity of over 96% (Figure 9), which is higher than the purity achieved by Kroupa et al. [26] and Kinčl et al. [27]. The only impurity in the acid compartment is sodium ions, whereas sulfate ions appear in the base compartment because a BPM can also leak small amounts of these ions [38]. As seen in Figure 9 below, high purity (>96%) of products was reached with the homogeneous SUEZ membranes. In contrast, Kinčl et al. could reach only 80% purity when using heterogeneous membranes [27].

### 3.3. Capital and Operational Expense Calculations

Based on the results above and the conditions assumed shown in Table 3, the CAPEX and OPEX for a 100 m^3^ h^−1^ BPED were calculated for a stream of 110 g L^−1^ Na_2_SO_4_. For projected acid production of 144 m^3^ h^−1^ at 1.0 N, and base production of 144 m^3^ h^−1^ at 1.0 N, an estimated 256 industrial BPED stacks (12,000 m^2^ total active membrane surface) would be required, operating at 1.4 kWh kg^−1^ of treated Na_2_SO_4_.

The CAPEX (These values might be varied in range ± 20%) comes to approximately EUR 20 M for all BPED equipment, excluding feed/product tanks, pre/post treatment and integration cost. The OPEX cost breakdown can be noted in the table below (Table 4) and assumes 8000 working hours per year, and maintenance includes membrane, stack and equipment maintenance. In this case, the costs for electricity make up almost half of all the operational costs. The total OPEX (These values might be varied in range ± 20%) comes to 14.2 M EUR/year, assuming pricing of 0.05 EUR/kWh for electricity and EUR 2 m^−3^ of DI-H_2_O. At 300 EUR/ton for 98% NaOH and 100 EUR/ton for 95% H_2_SO_4_ however, chemical consumption cost can be reduced by approximately 16.1 M EUR/year at production rates of 5.0 ton h^−1^ and 5.3 ton h^−1^ for NaOH and H_2_SO_4_, respectively, if the chemicals are reused at the site. Taking this into account, the process can be operated at a margin of approximately 1.8 M EUR/year. This cost off-set leads to an estimated payback time of 11 years. 

### 3.4. Sensitivity Analysis

Depending on market conditions and location, the average price for utilities and chemicals varies significantly and impacts OPEX values. For instance, an increase in electricity pricing from 0.05 to 0.09 EUR/kWh results in an increase from 14.2 to 19.2 M EUR/year for the OPEX, which in turn decreases the margin. Alternatively, higher chemical prices, i.e., NaOH costs from 300 to 400 EUR/ton, increase the cost-reduction margin by 24.5% from 16.1 to 20.1 M EUR/year and reduce the payback time to 3.4 years. Depending on the effluent stream process, membrane lifetime can vary significantly and will subsequently increase or decrease OPEX with regard to membrane maintenance. As stated previously, CAPEX and OPEX calculations are required for individual application cases.

The environmental footprint of the BPED should also be considered. The process itself does not cause any direct greenhouse gas emissions. However, the BPED technology is a highly intensive energy consumer leading into indirect effects on the environment from energy production [39]. The electricity should preferably be generated from renewable energy sources with a low CO_2_ footprint. Other environmental aspects can also be taken into consideration. The emissions arising from the transportation of purchased chemicals to the site can be avoided if the produced NaOH and H_2_SO_4_ can be used on site. The application of BPED is also potentially a cost-effective approach. The operational costs for purchased raw materials could be decreased and industrial sites currently disposing the stream could partially diminish disposal costs [40]. At the same time, environmental risks related to the disposal of Na_2_SO_4_ are mitigated. 

Both the study and the CAPEX/OPEX show promising results; for this reason, integration studies can be performed on BPED, e.g., into the leaching process of Ni during the refining process, one possibility being if integrated together with evaporators and reverse osmosis (RO) units, a zero liquid discharge setup can be achieved. In this theoretical process, the water recovered through both the RO and evaporator processes is used to dilute and maintain liquid levels for both the acid and base products. While RO and evaporators have relatively high operating costs (approximately EUR 5 m^−3^ for RO and EUR 30 m^−3^ for evaporators), their cost can be offset through government grants aimed at incentivizing zero liquid discharge operations. Similar flow diagrams could be integrated to other processes, such as battery material production. Recently, the potential of integrating BPED to treat Na_2_SO_4_-rich streams at an industrial level has been recognized in the battery sector [8].

## 4. Conclusions

We have shown that the BPED process can be used to effectively treat waste sodium sulfate effluent streams. Using state-of-the-art homogeneous membranes operating at 2.6 V per cell, 500 A m^−2^ and 41 °C, acid and base purity values exceeding 95% can be reached. These performance values are superior to the earlier performance reported for heterogeneous membranes. The primary impurities in the acid and base streams are sodium and sulfate ions, respectively, due to ion leakage through the BPM. Additionally, it was observed that the temperature increase led to an improvment in current efficiency. Varying the number of RUs from 5 to 10 had a negligible effect on the current efficiency, indicating that the shunt current in the short stack was insignificant. As increasing temperature improves both the voltage and current efficiency of the process, it is desirable to operate the system at the maximum temperature that the membrane can endure. During operation, the base production stream showed higher current efficiencies compared with the acid stream, which is attributed to the higher proton leakage through the AEM than hydroxide leakage through the CEM. BPED might serve in addition or as a replacement to existing processes such as evaporators and RO, with BPED showing specific energy consumption of 1.4 kWh kg^−1^ Na_2_SO_4_ treated and costing approximately 0.215 EUR/kg Na_2_SO_4_ treated, depending on conditions.

Overcoming some technical or economic barriers for the recovery of acid and base streams from effluent streams using BPED, including but not limited to both the BPED performance and pre-treatment, is required for a successful scale-up. For instance, effective pre-treatment of effluent streams to remove impurities can ensure long membrane lifetime, reducing the cost of operation. In addition, additional purification and concentration of the products might be required, depending on the given case. Further improvements regarding membrane performance and operating parameters can lead to decreased energy consumption and emphasize the techno-economic perspective. Future work would see the integration of a BPED system into an industrially relevant process, to determine the effects and issues associated with impurity build-up due to stream recycling in a closed loop. Furthermore, the integration of BPED with processes such as evaporators or RO should be investigated to determine whether a zero liquid discharge configuration is feasible.

The feasibility of integrating the BPED process to a given chemical process should always be considered case by case in detail, as the feed streams vary from site to site and prices of, e.g., chemicals and utilities depend on the location. Tailor-made solutions are often needed; however, the capital and operational expenditure calculations in this study give a good overview of the potential cost-effectiveness of the BPED process. With ever-increasing legislation, regulations regarding wastewater discharge and migration to zero liquid discharge processes, the demand for BPED technology is expected to increase.

## Figures and Tables

**Figure 1 membranes-11-00718-f001:**
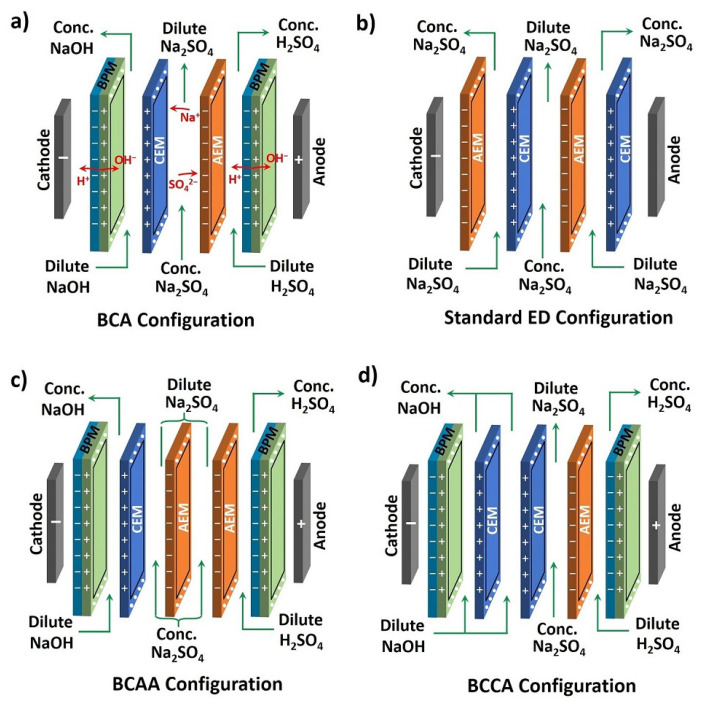
Different configurations of membrane electrodialysis process: (**a**) BCA configuration, (**b**) standard ED configuration, (**c**) BCAA configuration and (**d**) BCCA configuration.

**Figure 2 membranes-11-00718-f002:**
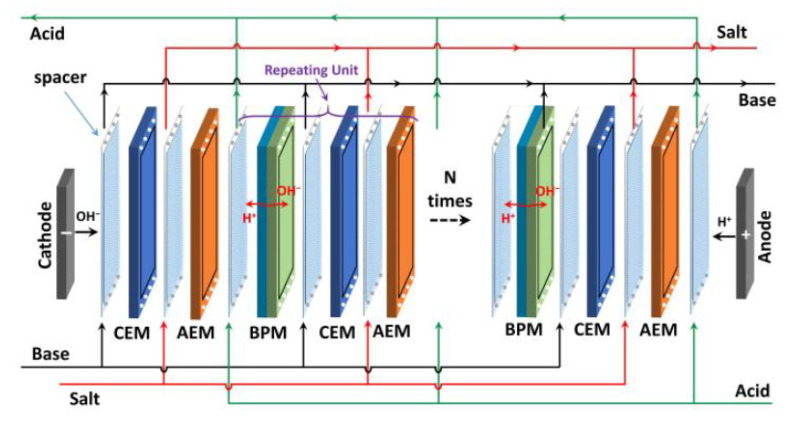
Schematic of BPED process with BCA as a repeating unit (RU).

**Figure 3 membranes-11-00718-f003:**
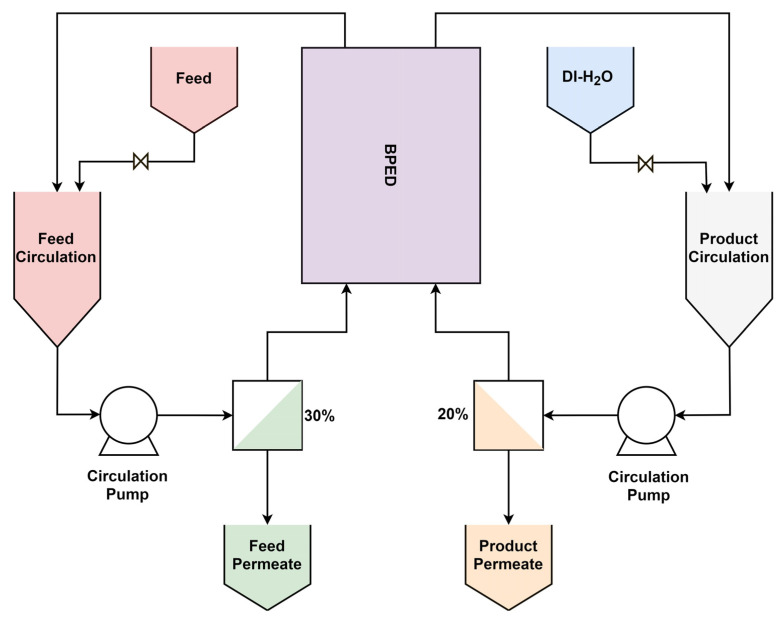
Schematic representation of the batch feed and bleed operation of the BPED process.

**Figure 4 membranes-11-00718-f004:**
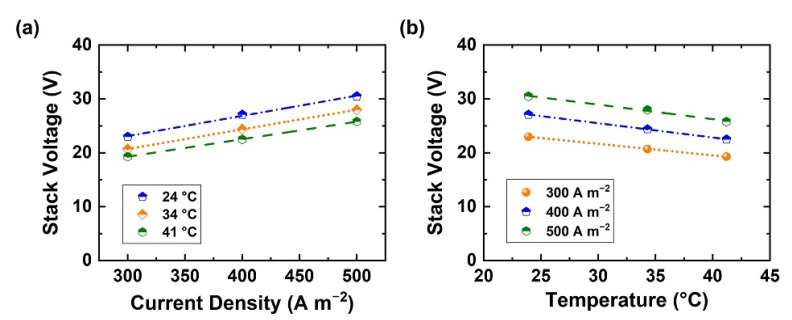
Cell voltage variation in Stack of 10 repeating units (**a**) at different current densities, and (**b**) at different temperature.

**Figure 5 membranes-11-00718-f005:**
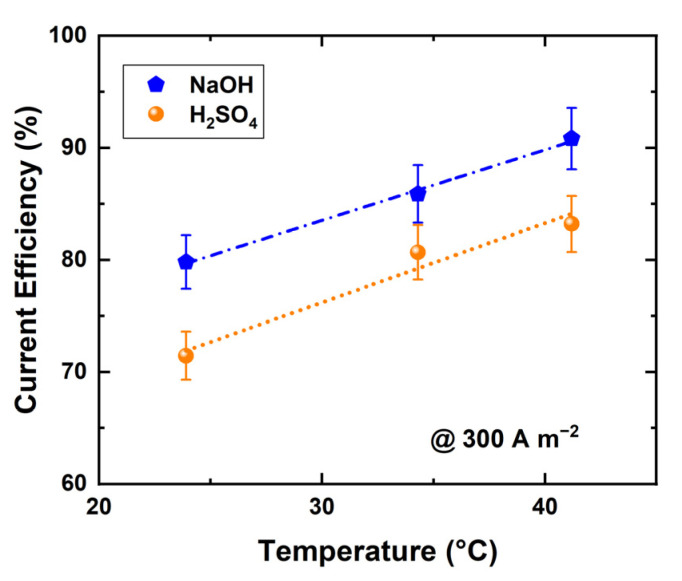
Current efficiency of BPED products in a stack of 10 RUs at different temperaTable 300. A m^−2^ (products concentrations were 0.5 ± 0.02 N).

**Figure 6 membranes-11-00718-f006:**
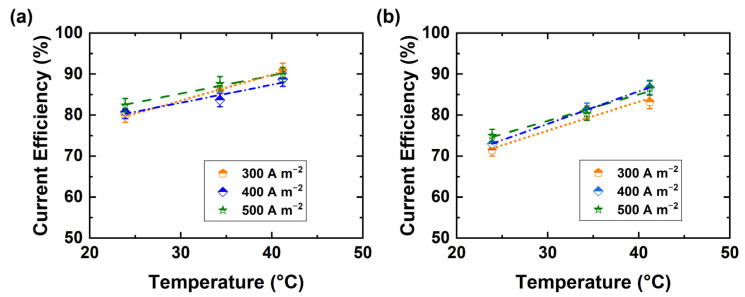
Current efficiency of BPED products in a stack of 10 repeating units at different current densities of (**a**) NaOH (0.5 ± 0.01 M), and (**b**) H_2_SO_4_ (0.25 ± 0.01 M).

**Figure 7 membranes-11-00718-f007:**
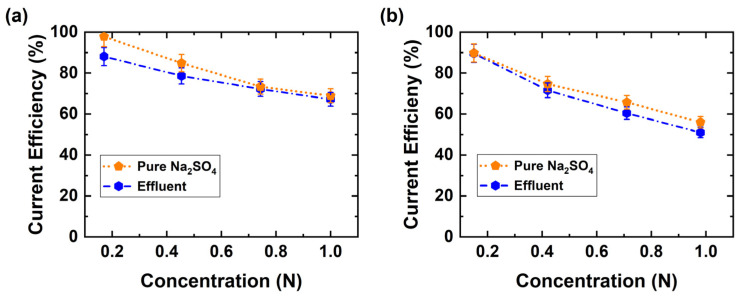
Current efficiency variation as a function of BPED products using pure Na_2_SO_4_ or the industrial effluent. (**a**) Base, (**b**) Acid (300 A m^−2^ at room temperature).

**Figure 8 membranes-11-00718-f008:**
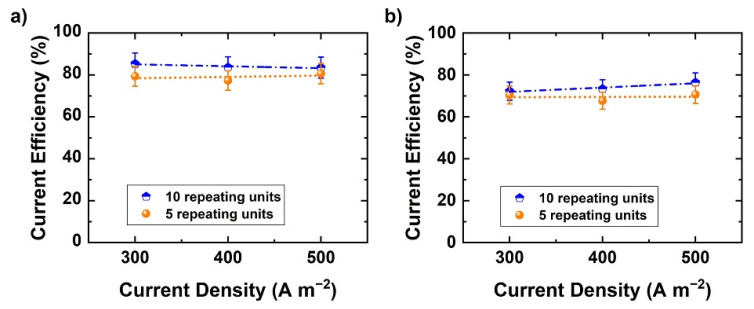
Current efficiency variation as a function of number of RUs at different current densities: (**a**) NaOH and (**b**) H_2_SO_4_.

**Figure 9 membranes-11-00718-f009:**
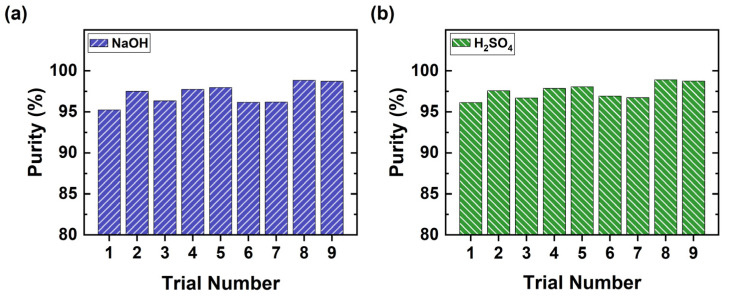
BPED of the effluent; purity of products (**a**) NaOH and (**b**) H_2_SO_4_.

**Table 1 membranes-11-00718-t001:** Membrane’s characteristics [34].

	Membrane	Area (cm^2^)	Thickness (μm)	WU (%)	IEC (meq/g)	*X_f_* (M) ^†^	*X_fp_* (M) ^††^
CEM	CR61P	10	580	44	2.20	1.23	3.1
AEM	AR103P	10	570	39	2.37	1.45	3.6
BPM	-	10	1150	-	-	-	-

**^†^** Membrane fixed charge. **^††^** Membrane fixed charge in pore water.

**Table 2 membranes-11-00718-t002:** Effluent composition.

Element	Ca	Mg	Si	NH_4_	TOC *	TSS **	Na_2_SO_4_
Value (mg L^−1^)	2.9	2.1	11.2	250	<100	<2000	110,000

* Total organic carbon. ** Total suspended solids.

**Table 3 membranes-11-00718-t003:** Operating condition.

Parameter	Value	Parameter	Value
Current (A m^−2^)	400	Base Efficiency (%)	80
Cell Voltage (V)	2.5	Acid Efficiency (%)	70
Temperature (°C)	40	Removal rate (%)	75

**Table 4 membranes-11-00718-t004:** OPEX estimations for operation of the BPED process.

Component	Operational Cost (M EUR/Year)	Fraction of Cost (%)
DI-H_2_O	4.3	30
Electricity	6.2	44
BPED Maintenance	3.3	23
Labor	0.4	3
Total	14.2	100

## Data Availability

Not applicable.

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
