# Peer review of "Bipolar Membrane Electrodialysis for Sulfate Recycling in the Metallurgical Industries"

_membranes, 2021, doi:10.3390/membranes11090718_

Round 1

Reviewer 1 Report

In the work “Bipolar Membrane Electrodialysis for Sulfate Recycling in the Metallurgical Industries”, the authors investigated the feasibility of using BPED to convert a sulfate-rich waste stream into a sulfuric acid stream and a NaOH stream. Several operating parameters, such as current density and temperature, were systematically varied to regress the optimal operation condition for the above BPED process. Techno-economic analysis (TEA) was then conducted to analyze the economic performance of the process. Overall, the paper has some practical values, and most of the results were well-discussed. Therefore, I would recommend this paper to be published after minor revision. Below are my comments that I hope the authors could address properly.

  • The introduction section is a little bit too long. I was wondering if the authors could move part of the contents (e.g., specific results from previous studies) between line 85 (Page 3) and line 121 (Page 4) to the results and discussions section?
  • In Table 2, the authors mentioned that the concentration of sodium sulfate is 110000 ppm, which is pretty high. I was wondering if diffusive ion transport would become non-trivial under this condition? Could the author please comment on the relative extent of diffusive ion transport to electric field-driven ion transport?
  • Page 6, line 196. Is the word “interphase” actually “interface”?
  • Page 6, line 199. The authors mentioned that the membrane resistance would decease with increasing temperature, which is not surprising. If the authors have actually measured the resistance, please include the values in the paper.
  • Page 7, line 215 through 217. Please add the units for the parameters defined here.
  • Page 7, line 221 through 223. The authors suggested that it is unclear why current efficiency increases. According to line 238 (page 8), a higher current efficiency corresponds to lower leakage of co-ions, i.e., the transport of target ion is more favorable. Given that the membranes were dense membranes in this work, I was wondering if the authors could try to explain the higher current efficiency at elevated temperature via the solution-diffusion model? Both the ion sorption and diffusion should be temperature dependent. Some of the recent works also formulated the ion transport in ED using this model (J. Membr. Sci. 2020, 597, 117645, Ind. Eng. Chem. Res. 2020, 59, 32, 14189–14206).
  • Page 9, line 272 through 275. The authors compared the purity of the product obtained in this work to a previous study. Is the type of membrane the dominating contributing factor to the product purity?
  • Page 10, section 3.3. I suggest the authors improve the TEA section, the current one is not convincing as many of the important economic parameters/considerations were not included here. Also, please add a table to summarize the parameters used in this section.

Author Response

We thank the reviewer for a detailed analysis and appropriate comments on our manuscript. We have done revisions on the manuscript and have replied to the best of our ability.

Abstract has been modified to reflect the observed current efficiency increase with temperature. And additional lines have been added to the conclusion.

Author Response

We thank the reviewer for a detailed analysis and appropriate comments on our manuscript. We have done revisions on the manuscript and have replied to the best of our ability.

Page 3 line 85 - page 4 line 121: Although the introduction part is a little longish, we intend to explain the previous work with different membrane configurations in the bipolar membrane electrodialysis using Figure 1. We ourselves reviewed it and concluded that this kind of literature study must be in introduction part. Results and discussion part is for our experimental analysis.

Table 2: No doubt, there is diffusion, because no ion exchange membrane is ideal, allowing the transport of co-ions, which is reflected in the value of the current efficiency. Yet, the concentration gradient across the membrane is not equal to the concentration of Na2SO4 (feed) over membrane thickness, because the feed compartment is separated from acid and base compartments. The transport problem is quite complicated because the system actually is quaternary. In our previous paper (Chem. Eng. J. Advances 2021, 100169, DOI: 10.1016/j.ceja.2021.100169) we analysed this problem in detail.

Page 6 line 196: Corrected.

Page 6 line 199: Membrane resistance values are not reported in this paper.

Page 7 Line 215-217: Units have been added.

Page 7 Line 221-223: A common understanding is that increasing temperature decreases membrane selectivity (Helfferich, Ion Exchange, p. 166). The fact that we observe an increased current efficiency would imply on the contrary. A possible reason is that the membrane swells somewhat, impeding salt diffusion, but this should be confirmed with diffusion experiments at varying temperatures. Hence, we are not able to crack this problem at the moment.

Page 9 Line 272-272: The comparison is made between heterogeneous membranes and homogeneous membranes, and hence it is probable that the reason for higher purity is the membrane. However, stack design and membrane configurations (already mentioned in the introduction itself) also plays a big role in determining product purity.

Page 10 section 3.3: Unfortunately expanding upon the considerations taken into the account to determine the OPEX and CAPEX is not possible at this time due to legal constraints.
